# Immunoregulatory role of the gut microbiota in inflammatory depression

Penghong Liu[1,2,3,6], Zhifen Liu ®[1,3,4,6], Jizhi Wang[1,2], Junyan Wang[1,2], Mingxue Gao[1,2], Yanyan Zhang[1,2], Chunxia Yang[1,3], Aixia Zhang[1,3], Gaizhi Li[1,2,3], Xinrong Li[1,3], Sha Liu[1,2,3], Lixin Liu[5], Ning Sun ®[1,2,3] ✉ & Kerang Zhang ®[1,2,3] ✉

Inflammatory depression is a treatment-resistant subtype of depression. A causal role of the gut microbiota as a source of low-grade inflammation remains unclear. Here, as part of an observational trial, we first analyze the gut microbiota composition in the stool, inflammatory factors and short-chain fatty acids (SCFAs) in plasma, and inflammatory and permeability markers in the intestinal mucosa of patients with inflammatory depression (ChiCTR1900025175). Gut microbiota of patients with inflammatory depression exhibits higher *Bacteroides* and lower *Clostridium*, with an increase in SCFA-producing species with abnormal butanoate metabolism. We then perform fecal microbiota transplantation (FMT) and probiotic supplementation in animal experiments to determine the causal role of the gut microbiota in inflammatory depression. After FMT, the gut microbiota of the inflammatory depression group shows increased peripheral and central inflammatory factors and intestinal mucosal permeability in recipient mice with depressive and anxiety-like behaviors. *Clostridium butyricum* administration normalizes the gut microbiota, decreases inflammatory factors, and displays antidepressant-like effects in a mouse model of inflammatory depression. These findings suggest that inflammatory processes derived from the gut microbiota can be involved in neuroinflammation of inflammatory depression.

Major depressive disorder (MDD) remains challenging to prevent and treat, mainly due to its highly heterogeneous features and multifactorial processes underlying the disease[1]. The neuroimmune hypothesis of depression was first proposed in 1995[2]. During the last few decades, several evidences have accumulated that inflammation plays an important role in the etiology and pathophysiology of MDD[3,4]. This type of depression is linked to increased pro-inflammatory cytokines such as interleukin 6 (IL-6), tumor necrosis factor alpha (TNF-α), C-reactive protein (CRP), and several other factors, and these inflammatory factors may help distinguish between inflammatory and non-

inflammatory depression[5-7]. Considering the stability, accuracy, and availability of various inflammatory indicators, the Centers for Disease Control and Prevention and American Heart Association (CDC/AHA) recommended CRP as an inflammatory marker in clinical and public health practice. Furthermore, the threshold of low-risk (<1.0 mg/L), average-risk (1.0–3.0 mg/L), and high-risk (>3.0 mg/L) correspond to the approximate tertiles of CRP in the adult population[8]. Similarly, the CRP concentration (>3.0 mg/L) has been used as a biomarker of inflammatory depression[9], which has unique clinical and biological characteristics[1]. In addition, patients with inflammatory depression

[1]Department of Psychiatry, First Hospital of Shanxi Medical University, Taiyuan 030001, PR China. [2]Shanxi Medical University, Taiyuan 030001, PR China. [3]Shanxi Key Laboratory of Artificial Intelligence Assisted Diagnosis and Treatment for Mental Disorder, First Hospital of Shanxi Medical University, Taiyuan 030001, PR China. [4]Key Laboratory of Cellular Physiology (Shanxi Medical University), Ministry of Education, Taiyuan 030001, PR China. [5]Experimental Center of Science and Research, First Hospital of Shanxi Medical University, Taiyuan 030001, PR China. [6]These authors contributed equally: Penghong Liu, Zhifen Liu ✉e-mail: sunning@sxmu.edu.cn; atomsxmu@vip.163.com

may represent a relatively treatment-resistant population[10]. For instance, the presence of inflammation in MDD is associated with a poorer response to first-line antidepressant therapies[11–13], accordingly, this subpopulation of patients is particularly challenging to treat. Therefore, it is crucial to explore the precise origin of inflammation and its pathogenic mechanisms to develop more effective treatments[14].

Studies suggest that the intestinal tract, especially disordered microbiota, could be a major source of inflammation that contributes to neurodegeneration[15,16]. The gut microbiota can directly modulate the immune system and contribute to the maintenance and breakdown of immune tolerance[17]. Furthermore, microbe-associated molecular patterns (MAMPs) expressed during dysbiosis are recognized by host pattern recognition receptors (PRRs), toll-like receptors (TLRs)[18], and nucleotide-binding oligomerization domain (NOD)-like receptors (NLRs) such as NLRP3 inflammasomes[19]. These receptors are the gateways to the innate immune system and are the first step in the cascade leading to cytokine production[20]. These cytokines increase intestinal permeability[21] and disrupt the blood-brain barrier (BBB). They can also activate microglia in the central nervous system (CNS)[22], leading to the secretion of pro-inflammatory cytokines that alter brain structure and function[23], further lead to clinical depression[24]. Gut microbiota have also been shown to produce a range of beneficial metabolites, such as short-chain fatty acids (SCFAs), which can affect immune and inflammatory responses[20]. SCFAs can affect intestinal mucosal immunity and the CNS immune system by crossing the BBB to modulate brain function[25]. Thus, gut microbiota may influence depressive symptoms by regulating the peripheral and central immune systems. Several studies have attempted to explore the "microbiota-gut-immune-brain axis" hypothesis in MDD[16,26], but their results are inconsistent due to the high heterogeneity of depression. Therefore, it is necessary to elucidate this hypothesis by considering the inflammatory subtypes of depression. In conclusion, disturbed gut microbiota may be the source of low-grade inflammation. As of now, the gut microbiota's features and its role of immunoregulation in this subtype of depression remains uncertain.

To clarify the gut microbiota composition and the mechanisms involved in inflammatory depression, we performed experiments using human and mouse samples. First, we analyzed the gut microbiota, intestinal permeability, SCFA concentration, and inflammatory cytokine concentration in patients with inflammatory depression. We found pro-inflammatory genera was increased and SCFA-producing genera was decreased in inflammatory depression patients. Then, we performed fecal microbiota transplantation (FMT) and probiotic supplementation, and identified that disordered gut microbiota can induce inflammatory depression by activating Toll-like receptor 4 (TLR-4)/nuclear factor kappa-B (NF-κB) and NLRP3-mediated inflammatory cascades and the SCFA-producing gut microbes can regulate inflammatory responses and improve depressive symptoms. The results lay a foundation for potential prevention/intervention/treatment of inflammatory depression through gut microbiota and metabolome alteration.

## Results
### Clinical study
#### Part I: Human study in subjects with MDD and healthy controls (HCs)
**Clinical characteristics, inflammatory factors, and SCFAs in MDD and HCs.** Patients with MDD and HCs were similar in age, sex, and body mass index (BMI) ($P > 0.05$). The patients in this cohort had no other somatic conditions and were not overweight. Patients with MDD had significantly fewer years of education than the HCs ($P < 0.001$). The MDD group had higher total Hamilton Rating Scale for Depression (HAMD-17) and Hamilton Anxiety Rating Scale (HAMA) scores than the HC group ($P < 0.001$). We detected higher level of hs-CRP in the MDD

group than that in the HC group ($P < 0.05$); however, IL-1β, IL-6, IL-10, and TNF-α concentration showed no significant differences between the two groups ($P > 0.05$). The MDD group had a significantly lower propionic and butyric acid levels than the HCs ($P < 0.05$). However, acetic acid, isobutyric acid, valeric acid, and caproic acid levels were comparable between the two groups ($P > 0.05$) (Table 1).

### Gut microbiota composition between MDD patients and HCs
Alpha-diversity analysis showed that the Chao index, Faith's Phylogenetic Diversity (PD), and the number of observed species were higher in patients with MDD than those in HCs. However, Good's coverage was lower in patients with MDD ($P < 0.05$) (Fig. 1A). Beta diversity analysis demonstrated that the difference between the groups was larger than the differences within the groups, with permutational multivariate ANOVA (PERMANOVA) analysis based on Jaccard dissimilarity ($P < 0.001$) (Fig. 1B). A linear discriminant analysis effect size (LEfSe) test was used to investigate microbiota discrepancies between patients with MDD and HCs. At the family level, the relative abundances of Micromonosporaceae and Rhodospirillaceae were significantly higher in patients with MDD; however, the abundances of Clostridiaceae, Peptostreptococcaceae, Pasteurellaceae, and Turicibacteraceae were significantly higher in HCs. At the genus level, the relative abundance of *Adlercreutzia* was significantly higher in patients with MDD; however, the abundances of *Clostridium, Roseburia, Haemophilus, SMB53,* and *Turicibacter* were much higher in HCs (Fig. 1C).

**Table 1 | Clinical characteristics, inflammatory parameters and SCFAs in MDD and HCs**

| Variable | MDD (n = 85) | HCs (n = 85) | t/χ² | P |
|---|---|---|---|---|
| Sex(Male/Female) | 39/46 | 38/47 | 0.024 | 0.878[a] |
| Age (years) | 24.32 ± 8.78 | 24.08 ± 3.55 | 0.229 | 0.819[b] |
| Education years | 12.89 ± 3.28 | 16.70 ± 2.26 | 8.798 | 0.001[b*] |
| BMI | 21.786 ± 3.54 | 22.08 ± 2.53 | 0.669 | 0.504[b] |
| HAMD-17 | 26.31 ± 5.96 | 2.16 ± 4.09 | 30.817 | 0.001[b*] |
| HAMA | 18.74 ± 5.83 | 1.99 ± 3.07 | 21.425 | 0.001[b*] |
| hs-CRP (ng/ml) | 115.13 ± 23.86 | 98.36 ± 22.98 | 4.465 | 0.001[b*] |
| IL-1β (pg/ml) | 196.54 ± 40.00 | 198.92 ± 41.01 | 0.445 | 0.657[b] |
| IL-6 (pg/ml) | 117.76 ± 25.95 | 122.74 ± 23.40 | 0.922 | 0.359[b] |
| IL-10 (pg/ml) | 149.47 ± 35.68 | 149.08 ± 39.53 | 0.106 | 0.915[b] |
| TNF-α (pg/ml) | 476.85 ± 123.93 | 473.89 ± 106.85 | 0.071 | 0.908[b] |
| Acetic acid (µg/mL) | 2.02 ± 2.05 | 3.41 ± 2.39 | 1.365 | 0.174[b] |
| Propionic acid (µg/mL) | 0.20 ± 0.23 | 0.37 ± 0.24 | 4.362 | 0.001[b*] |
| Butyric acid (µg/mL) | 0.031 ± 0.046 | 0.056 ± 0.047 | 3.258 | 0.001[b*] |
| Isobutyric acid (µg/mL) | 0.015 ± 0.020 | 0.012 ± 0.016 | 0.753 | 0.453[b] |
| Isovaleric acid (µg/mL) | 0.073 ± 0.106 | 0.079 ± 0.127 | 0.294 | 0.770[b] |
| Valeric acid (µg/mL) | 0.005 ± 0.031 | 0.011 ± 0.057 | 0.729 | 0.467[b] |
| Caproic.acid (µg/mL) | 0.019 ± 0.033 | 0.087 ± 0.368 | 1.605 | 0.111[b] |

Data are presented as mean ± standard deviation (SD) for continuous variables.
All statistical tests are two-sided.
*SCFAs* short chain fatty acids, *MDD* major depressive disorder, *HC* health control, *BMI* Body Mass Index, *HAMD* Hamilton's Depression Scale, *hs-CRP* high-sensitivity C-reactive protein, *IL-1β* interleukin 1β, *IL-6* interleukin 6, *IL-10* interleukin 10, *TNF-α* tumor necrosis factor α.
*significant difference.
[a]P value for chi-square test.
[b]P values for two-sample t-test.

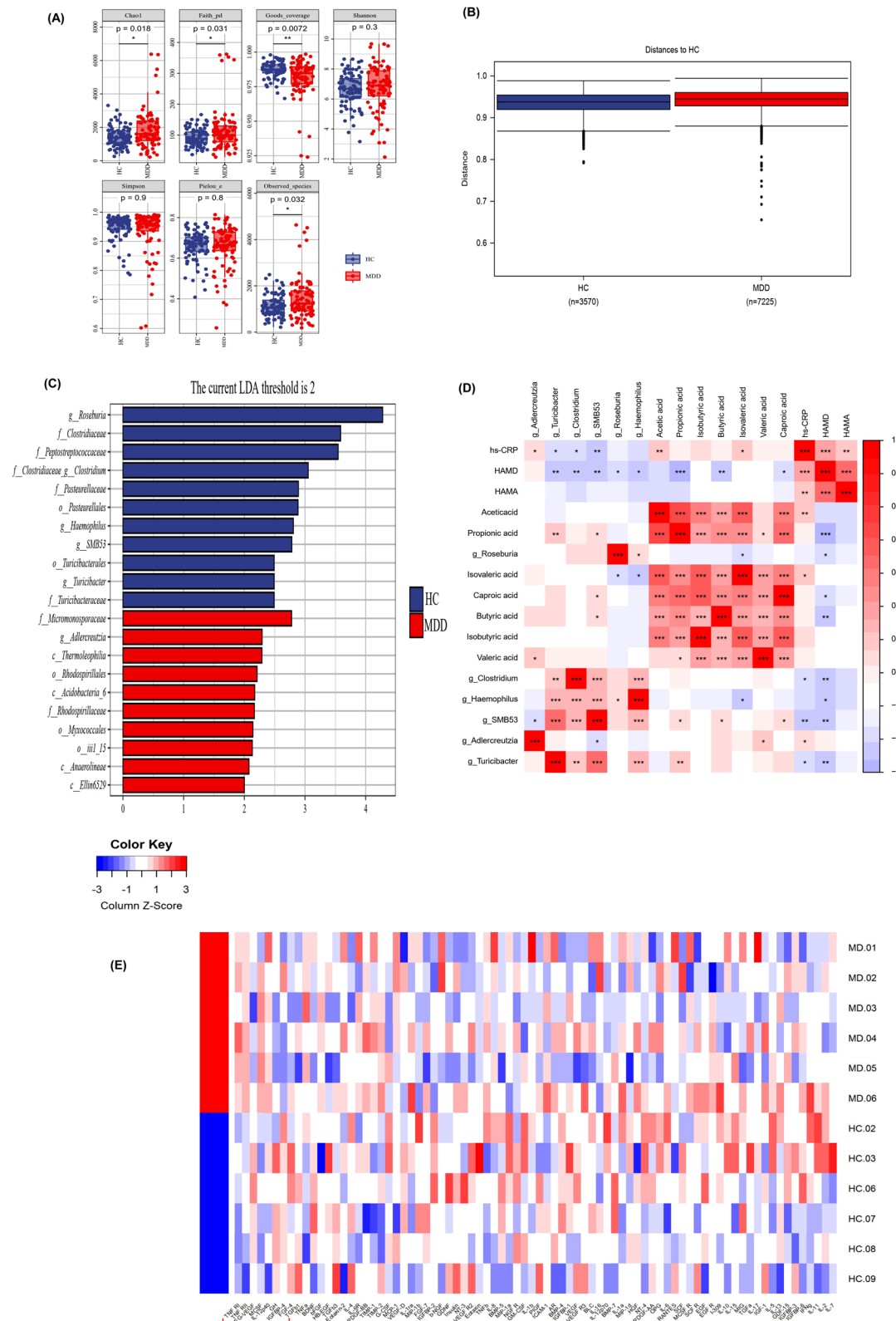

### Relationship between microbial biomarkers, hs-CRP, SCFAs, and depressive symptoms

To clarify the relationship between gut microbiota, hs-CRP, SCFAs, and depressive symptoms, we performed a correlation analysis and found that the relative abundance of *Clostridium, Roseburia, Haemophilus,* *SMBS3,* and *Turicibacter* was positively correlated with propionic acid and butyric acid and negatively correlated with hs-CRP and total HAMD-17 scores in all subjects (*P* < 0.05). The hs-CRP level also positively correlated with the total HAMD-17 and HAM-A scores (*P* < 0.05) (Fig. 1D).

**Fig. 1 | Gut microbial composition, short chain fatty acids (SCFAs) and inflammatory indicators of patients with MDD. A** Alpha-diversity analysis exposed that the Chao index, Faith's Phylogenetic Diversity (PD), Observed species were higher in MDD patients (*n* = 85) than in HCs (*n* = 85), The Good's coverage was lower in MDD patients (*P* < 0.05). Two-sample T test was used. Data are presented as mean values with SD. Box plots indicate median and interquartile range. **B** Beta diversity analysis uncovered that the difference between MDD patients (*n* = 85) and in HCs (*n* = 85) was larger than the difference within group by the permanova analysis based on Jaccard dissimilarity (*P* = 0.0006). Permutation test was used. Data are presented as mean values with SD. Box plots indicate median and inter-quartile range. The upper and lower whiskers indicate minima and maxima. **C** A linear discriminant analysis (LDA) effect size (LEfSe) showed that at the family level, the relative abundance of Micromonosporaceae and Rhodospirillaceae was significantly higher in MDD patients, however, the amount of Clostridiaceae, Peptostreptococcaceae, Pasteurellaceaewas and Turicibacteraceae significantly higher in HCs. At the genus level, the relative abundance of *Adlercreutzia* was significantly higher in MDD patients, however, the abundance of *Clostridium, Roseburia, Haemophilus, SMB53*, and *Turicibacter* were much higher in HCs. Blue and red colors represent HCs and MDD, respectively. **D** The correlation analysis were performed among different genus of gut bacteria, short chain fatty acids (SCFAs), hs-CRP and

severity of depression and anxiety. It was found the relative abundance of *Clostridium, Roseburia, Haemophilus, SMBS3*, and *Turicibacter* were the level of Acetic acid, Propionic acid and Butyric acid were positively correlated with Propionic acid and Butyric acid, and negatively correlated with hs-CRP and the total score HAMD-17 in all subjects (*P* < 0.05). The hs-CRP was positively correlated with the total score HAMD-17 and HAMA (*P* < 0.05). Pearson correlation analyses were implemented with FDR correction. The red and blue of the table represent positive and negative correlations respectively. *P*-value is marked as follows: ***P* ≤ 0.001; ***P* ≤ 0.01; **P* ≤ 0.05. **E** Different cytokine in intestinal mucosa between MDD (*n* = 6) and HCs (*n* = 6). The antibody cytokine array consisting of 80 inflammatory factors was used to screen for differential intestinal mucosal inflammatory factors between MDD and HCs. The heatmap shows that tumor necrosis factor alpha-R1 (TNF-R1), TNF-R2, Macrophage Colony Stimulating Factor (MCSF) and interleukin 12 (IL-12) are overexpressed, however, Growth Hormone (GH), Fibroblast Growth Factors 4 (FGF-4), Transforming Growth Factor 1β (TGF-1β) and Endocrine Gland-derived Vascular Endothelial Growth Factor (EG-VEGF) that involved in intestinal mucosal repair are underexpressed in MDD patients compared to HCs. The data was analyzed by moderated t-statistics and corrected by the Benjamini–Hochberg method. All statistical tests are two-sided. Source data are provided as a Source Data file.

## Inflammatory parameters and permeability biomarkers for intestinal mucosa in patients with MDD and HCs

To explore the effect of the disturbed gut microbiota on the intestinal mucosa, intestinal mucosal samples from 12 subjects (MDD = 6, HC = 6) (Supplementary Table S1) were screened using a glass slide-based antibody cytokine array containing 80 inflammatory proteins (Ray-Biotech, GSH-INF-3, GSH-GF-1). The heatmap shows that some cytokine proteins involved in inflammatory activation, such as TNF-R1, TNF-R2, macrophage colony-stimulating factor (MCSF), and IL-12, were overexpressed. However, growth hormone (GH), fibroblast growth factor 4 (FGF-4), transforming growth factor 1β (TGF-1β), and endocrine gland-derived vascular endothelial growth factor (EG-VEGF), all of which are involved in intestinal mucosal repair, are underexpressed in patients with MDD compared to HCs (Fig. 1E).

To further clarify the inflammatory pathway and quantify the intestinal permeability markers, Enzyme-Linked Immunosorbent Assay (ELISA) was used. TLR-4, NF-κB, and NLRP3 levels increased in the MDD group (*P* < 0.05) (Fig. 2A). Levels of permeability biomarkers, such as Claudin-1, ZO-1, and Occludin, were decreased (*P* < 0.05) (Supplementary Table S1) (Fig. 2B); this was confirmed by immunohistochemistry.

Correlation analyses showed that the relative abundance of *Clostridium* was negatively correlated with the levels of MCSF and hs-CRP, and inflammatory factors were negatively correlated with permeability biomarkers (Claudin-1, ZO-1, Occludin) and SCFAs (acetic acid, propionic acid, and butyric acid); they were positively correlated with HAMD-17 total score. The butyric acid level was positively correlated with occludin and negatively correlated with TLR-4, NF-κB, NLRP3, TNFR2, and HAMD-17. Claudin-1 was positively correlated with the intestinal mucosal repair markers, such as GH and EG-VEGF (Fig. 2C, Supplementary Fig. S2).

**Part II: Human study in subjects with inflammatory depression, non-inflammatory depression, and HCs.** Not all depressions are associated with an inflammatory response. It is estimated that inflammatory depression accounts for only 30% of all depressions. Since inflammatory depression has a poor response to first-line anti-depressant therapies, it is necessary to study the pathogenesis of different subtypes of depression. We further divided patients with MDD into an inflammatory depression group (*n* = 42) and a non-inflammatory depression group (*n* = 43) based on plasma hs-CRP levels. The cut-off value[8] was the second tertile (66.7%) of hs-CRP in the peripheral blood of HCs (*n* = 128), with a value of 110.67 (Supplementary Table S3).

## Gut microbiota features in inflammatory depression

To clarify the characteristics of the gut microbiota in inflammatory depression, LEfSe analysis was performed among the three groups. Compared with non-inflammatory depression and HCs, the relative abundances of pro-inflammatory bacteria, such as Bacteroidaceae and *Bacteroides* were significantly higher. It was also found that the relative abundances of anti-inflammatory bacteria, such as Clostridiaceae and *Clostridium*, which can produce SCFAs, were lower in patients with inflammatory depression (Fig. 3A). To determine the biomarkers between patients with inflammatory depression and HCs, and between inflammatory depression and non-inflammatory depression at the genus level, a receiver operating characteristic (ROC) curve was constructed by combining all different genera. ROC curve analysis showed that the AUC was 81.73% and 79.16%, respectively (Fig. 3B).

To further identify the specific microbiota at the species level and guide the treatment for inflammatory depression, stool samples from 20 patients with inflammatory depression and 20 randomly selected HCs were analyzed using shotgun metagenomic sequencing. LEfSe analysis revealed that the relative abundances of 43 species were significantly lower in the inflammatory depression group, and nine bacterial species were enriched in the inflammatory depression group (Fig. 3C). To create accurate diagnostic models for inflammatory depression, ROC curves were constructed by combining all the different species, and the AUC was found to be 100% (Fig. 3D).

The Kyoto Encyclopedia of Genes and Genomes (KEGG) orthology (KO) database was used to analyze functional gene sequences between the two groups. We found that the expression of K00244 (fumarate reductase A) was significantly decreased in the inflammatory depression group. K00244 was annotated into the KEGG pathway maps, and we found that butanoate metabolism (map00650) was abnormal in these samples (Supplementary Fig. S2). Correlation analysis found that the K00244 abundance was significantly and negatively correlated with the HAMD-17 scores (r = −0.439, *P* = 0.005), and that it was positively correlated with the level of butyric acid (r = 0.374, *P* = 0.023) (Fig. 3E). Some decreased species (*Clostridium_butyricum, Clostridium_sp.CAG_343, Clostridium_sp.CAG_417, Faecalibacterium_sp. AF27-11BH*, and *Eubacterium_sp.AF36-5BH*) in inflammatory depression correlated with hs-CRP, SCFAs, K00244, and depressive symptoms. Moreover, *Mogibacterium diversum* levels were positively correlated with hs-CRP and depressive symptoms (Fig. 3F).

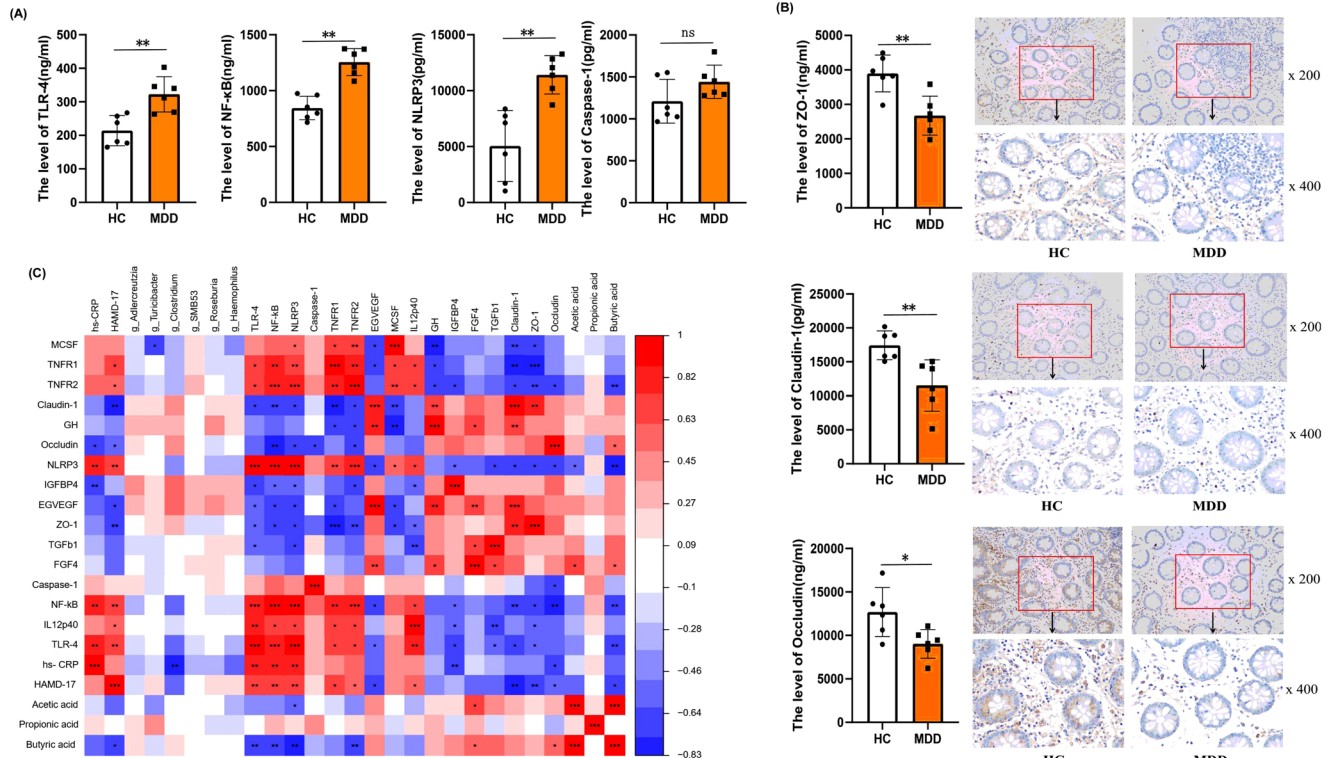

**Fig. 2 | Differences in intestinal mucosal inflammatory factors and permeability biomakers between MDD (*n* = 6) and HCs (*n* = 6). A** ELISA was used to measure related markers and found TLR-4 (*P* = 0.0033), NF-κB (*P* = 0.0014), NLRP3 (*P* = 0.0017) were increased in MDD group. Two-sample T test was used. Data are presented as mean values with SD. ns indicates non-significant. Scatter plot indicate median and error range. The upper and lower whiskers indicate median±SD. **B** The permeability biomakers such as Claudin-1 (*P* = 0.0070), ZO-1 (*P* = 0.0030), Occludin (*P* = 0.0210) were decreased that was confirmed by immunohistochemistry (×200 and ×400 magnificent). Two-sample T test was used. Data are presented as mean values with SD. Scatter plot indicate median and error range. The upper and lower whiskers indicate median ± SD. **C** The correlation analysis were performed and

found the relative abundance of *Clostridium* were negatively correlated with the level of MCSF and hs-CRP; The inflammatory factor were negatively correlated with the permeability biomarkers (Claudin-1, ZO-1, Occludin) and SCFAs (Acetic acid, Propionic acid, Butyric acid); and positively correlated with the total score of HAMD-17. The Butyric acid was positively correlated with Occludin and negatively correlated with TLR-4, NF-κB, NLRP3, TNFR2 and HAMD-17. The Claudin-1 was positively correlated with intestinal mucosal repair markers such as GH and EG-VEGF. Pearson correlation analyses were implemented with FDR correction. The red and blue of the table represent positive and negative correlations respectively. *P*-value is marked as follows: ***\*P* ≤ 0.001; **\*P* ≤ 0.01; *\*P* ≤ 0.05. All statistical tests are two-sided. Source data are provided as a Source Data File.

## The inflammatory factors and permeability biomarkers of intestinal mucosa in inflammatory depression

Compared with the non-inflammatory depression group (*n* = 3), the inflammatory group showed increased inflammatory factors: TLR-4, NF-κB, NLRP3, Caspase-1, TNF-RI, TNF-RII, and MCSF. Moreover, the permeability of biomarkers ZO-1 and occludin were decreased in the intestinal mucosa of inflammatory depression (*n* = 3). However, only the differences among NF-κB, TNF-RII, and occludin were statistically significant (*P* < 0.05) (Fig. 3G).

## Animal study

### Part I: The mouse model of inflammatory depression was established by FMT

**Gut microbiome transplantation from patients with inflammatory depression induces depressive-like behaviors in recipient mice.** To determine whether depression-related behavioral phenotypes were linked to disturbed gut microbiota, we performed FMT experiments (Fig. 4A). After FMT, the mice in the high inflammation group consumed less sucrose in the sucrose preference test (SPT), indicating that the mice showed more anhedonia-like behavior (*P* < 0.05). In the open-field test (OFT), mice in the high inflammatory group showed decreased activity (less total distance traveled) and increased anxiety (reduced travel in the exposed center region away from the walls) compared to the low-inflammatory group, HC group, and blank group (*P* < 0.05). Similarly, the duration of immobility in the tail suspension test (TST) increased in the

high-inflammatory group compared to that in the HC group (*P* = 0.061) and blank group (*P* = 0.070), suggesting increased depressive-like behavior (despair). The weight of mice in the high inflammatory group was significantly lower than that in the low-inflammatory group, HC group, and blank group (*P* < 0.05), and the weight change of mice in the high inflammatory group was smaller than that in the other groups (*P* < 0.05) (Fig. 4B, C). To determine the effect of fluid intake on body weight, we compared the total fluid intake of the mice during behavioral testing. However, there was no statistically significant difference between the groups (*P* > 0.05). Collectively, these behavioral tests showed that mice transplanted with the microbiota of patients with inflammatory depression displayed lower weight, smaller weight change, decreased activity, increased anxiety, and depressive-like behaviors.

## Dysbiosis of gut microbiota in the high-inflammatory group

To determine whether the discriminative microbial markers characteristic of inflammatory depression successfully colonized the high inflammation group, we characterized the gut microbial composition.

Alpha-diversity analysis showed that the Simpson index was lower in the high inflammatory group than that in the low-inflammatory group, HC group, and blank group (*P* < 0.05) (Fig. 4D1). Beta diversity analysis revealed a notable difference in the bacterial community composition between the high-inflammatory group and other groups, as determined by Jaccard dissimilarity

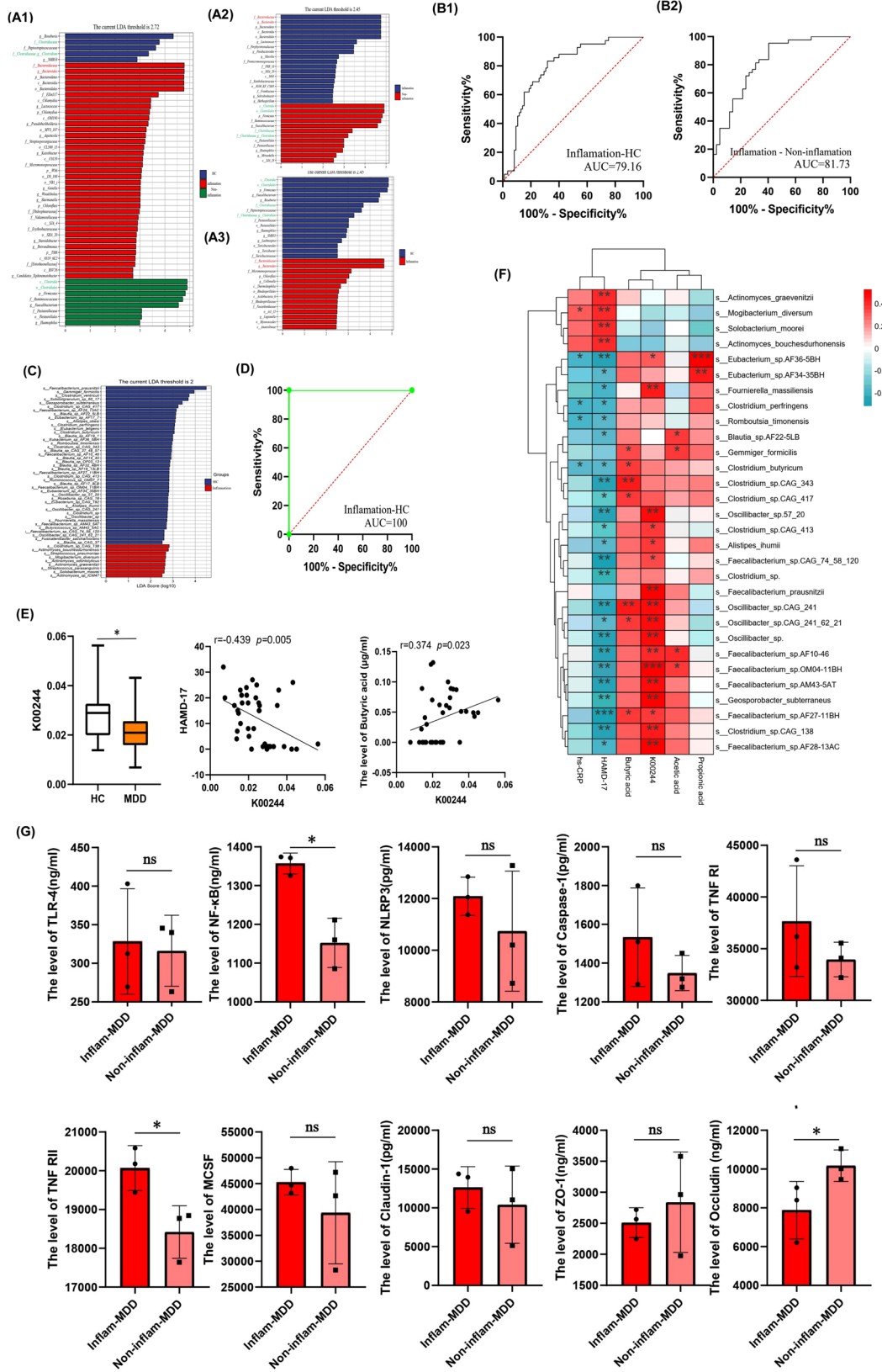

calculations (Fig. 4D2). LEfSe analysis was used to investigate the microbiota discrepancies among the four groups and revealed that the relative abundances of Bacteroidaceae and *Bacteroides* were significantly higher in the high-inflammatory group; however, the amounts of Clostridiales and *Clostridia* were significantly higher in the HC group (Fig. 4D3).

## Elevated inflammatory factors and permeability in the intestinal mucosa in the high inflammatory group

Disturbed microbiota can affect the immune and inflammatory pathways, leading to immune activation and inflammation of the intestinal mucosa, further damaging the intestinal barrier[27]. Therefore, we investigated the effects of gut microbiota in inflammation by qPCR and

**Fig. 3 | Gut microbiota characteristics, intestinal mucosal inflammatory factors and permeability biomakers of inflammatory depression. A** To clarify the characteristics of gut microbiota in inflammatory depression, The LEfSe analysis was performed among three groups and found that compared with Non-inflammatory depression and HCs, the relative abundance of Bacteroidaceae and Bacteroides were significantly higher, Clostridiaceae and Clostridium were lower in Inflammatory depression patientsl. **B** To determine the biomarkers for discriminating between inflammatory depression and HC, between inflammatory depression and Non-inflammatory depression at the genus level, the receiver operating characteristic (ROC) curve were made to by combining all different genus. The ROC analysis showed that the AUC was 81.73% and 79.16% separately. **C** To further identify the specific microbiota at species level and guide treatment, the shotgun metagenomic sequencing was used for gut microbiota of inflammatory depression (*n* = 20) and HCs (*n* = 20). We performed LEfSe analysis and found at the species level, the relative abundance of 43 species was significantly lower in the inflammatory depression group, and 9 bacterial species were enriched in Inflammatory depression. **D** To make accurate diagnostic models for inflammatory depression, ROC curve were made to by combining all different species and found the AUC was 100%. **E** The different Kyoto Encyclopedia of Genes and Genomes (KEGG) orthology (KO) analysis found the expression of K00244 (fumarate reductase A, an enzyme involved in the butanoate metabolism) was significantly decreased in Inflammatory depression group (*n* = 20). Two-sample T test was used. Data are presented as mean values with SD. Box plots indicate median and interquartile range. The upper and lower whiskers indicate minima and maxima. The correlation analysis found the K00244 abundance was significant negative correlated with the HAMD-17 scores (r = −0.439, *P* = 0.005), and positive correlated with the level of butyric acid (r = 0.374, *P* = 0.023). Pearson correlation analyses were implemented. **F** The correlation analysis showed that the relative abundance of abnormal gut microbiota were associated with hs-CRP, SCFAs, K00244, HAMD-17. Red and blue color represent positive correlation and negative correlation, respectively. Pearson correlation analyses were implemented with FDR correction. *P*-value is marked as follows: ***P ≤ 0.001; **P ≤ 0.01; *P ≤ 0.05. **G** The inflammatory factors and permeability biomakers of intestinal mucosa in inflammatory depression(*n* = 3). Compared with Non-inflammatory depression group (*n* = 3), the inflammatory factor TLR-4 (*P* = 0.812), NF-κB (*P* = 0.018), NLRP3 (*P* = 0.392), Caspase-1 (*P* = 0.302), TNF-RI (*P* = 0.318), TNF-RII (*P* = 0.033), MCSF (*P* = 0.371), were increased and the permeability biomakers such as ZO-1 (*P* = 0.537), Occludin (*P* = 0.048) were decreased in intestinal mucosa of inflammatory depression. Two-sample T test was used. Data are presented as mean values with SD. Scatter plot indicate median and error range. The upper and lower whiskers indicate median± SD. *P*-value is marked as follows: *P ≤ 0.05. ns indicates non-significant. All statistical tests are two-sided. Source data are provided as a Source Data File.

the permeability of the intestinal mucosa using immunohistochemistry. We found that the expression of inflammatory factors, such as TLR-4 and NLRP3 (Fig. 5A), was elevated in the intestinal mucosa of the high inflammatory group compared to that in the low-inflammatory group, HC group, and blank group (*P* < 0.05). Furthermore, we found that the amount and area density of permeability biomarkers, such as Claudin-1 and ZO-1 (Fig. 5B), were decreased in the high inflammation group (*P* < 0.05). The area density of occludin tended to decrease in the high inflammatory group, but there was no statistically significant difference between the high and low inflammation groups (*P* > 0.05).

### Elevated inflammatory factors in the serum and brain of mice in the high-inflammatory group

Leaky gut can lead to bacterial translocation across the intestinal barrier into the circulatory system[21]. It can also activate inflammatory responses for circulation that can disrupt the BBB. Moreover, it can activate the microglia of the CNS and release pro-inflammatory cytokines[22]. We investigated inflammatory factors in the serum and brains of mice and found that the concentration of hs-CRP in serum (Fig. 5A) and the expression of TLR-4, NF-κB, NLRP3, Caspase-1, and IL-1β in the brain (Fig. 5C) were increased in the high inflammatory group compared to the low-inflammatory group, HC group, and blank group (*P* < 0.05). Furthermore, we assessed the number and morphology of microglia in the hippocampus using immunofluorescence and found that the area density of iba-1 and the branches of the microglia increased in the high inflammatory group (*P* < 0.05) (Fig. 5D).

### Altered gut microbiota are related to inflammatory markers, permeability markers and depressive-like behaviors in all mice

To identify the correlation between altered gut microbiota, inflammatory markers, permeability markers, and depressive-like behaviors, we performed correlation analysis and found that Bacteroidaceae and *Bacteroides* were positively correlated with inflammatory markers and depressive-like behaviors, and negatively correlated with permeability markers. However, Clostridiales and *Clostridia* were negatively correlated with inflammatory markers and depressive-like behaviors, and positively correlated with permeability markers. Inflammatory markers were negatively correlated with permeability markers and positively correlated with depression-like behaviors. We also found that Bacteroidaceae and *Bacteroides* negatively correlated with Clostridiales and *Clostridia* respectively (Supplementary Fig. S3).

## Part II

### Effects of probiotics *Clostridium butyricum* (CB) on the mouse model of inflammatory depression.

Overall, we found that the abundance of pro-inflammatory bacteria increased and that of anti-inflammatory bacteria decreased in the high inflammatory mice. To further examine the relationship between gut microbiota and depressive symptoms in the high-inflammatory mice, we supplemented the high-inflammatory mice with butyric-producing anti-inflammatory bacteria (CB) and found through alpha-diversity analysis in the CB group that the Simpson index increased (Fig. 6A). We also found through beta diversity analysis that the bacterial community composition was notably different between the high inflammatory group and the CB group (Fig. 6B). Furthermore, LEfSe analysis showed that the abundance of Clostridia and *Clostridiales* increased, and that of Bacteroidaceae and *Bacteroides* decreased in the CB group (Fig. 6C).

Compared with the high inflammatory group, the expression of the inflammatory factors TLR-4 and NLRP3 in the intestinal mucosa, the concentration of hs-CRP in serum (Fig. 7A), and the expression of TLR-4, NF-κB, NLRP3, Caspase-1, and IL-1β in the brain (Fig. 7C) decreased in the CB group (*P* < 0.05). We found that the permeability biomarkers Claudin-1 and ZO-1 increased in the CB group (*P* < 0.05); while occludin tended to increase. However, the difference was not statistically significant (Fig. 7B). The number and branches of microglia in the hippocampus decreased in the CB group (*P* < 0.05) (Fig. 7D).

Furthermore, depression-like behaviors were assessed using behavioral tests. Mice in the CB group consumed more sucrose in the SPF than did the high-inflammatory group or normal saline (NS) group (*P* < 0.05). In the OFT, mice in the CB group showed increased activity (longer total distance traveled) and decreased anxiety (more travel in the exposed central region away from the walls) (*P* < 0.05). Similarly, the duration of immobility in the TST was decreased in the CB group (*P* < 0.05). The weight of the mice significantly increased, and the weight change was greater in the CB group (*P* < 0.05) (Fig. 8).

## Discussion

In this clinical study, we found that, compared to HCs, the MDD group had significantly higher hs-CRP and lower propionic acid and butyric acid levels. Furthermore, we found that patients with MDD showed different gut microbial diversity and composition, in turn demonstrating that neuroinflammation is involved in the onset and development of depression[5,28,29]. We propose that the gut microbiota triggers neuroinflammation and immune activation in the brain[16,28]. Several studies show that manipulation of the intestinal microbiota

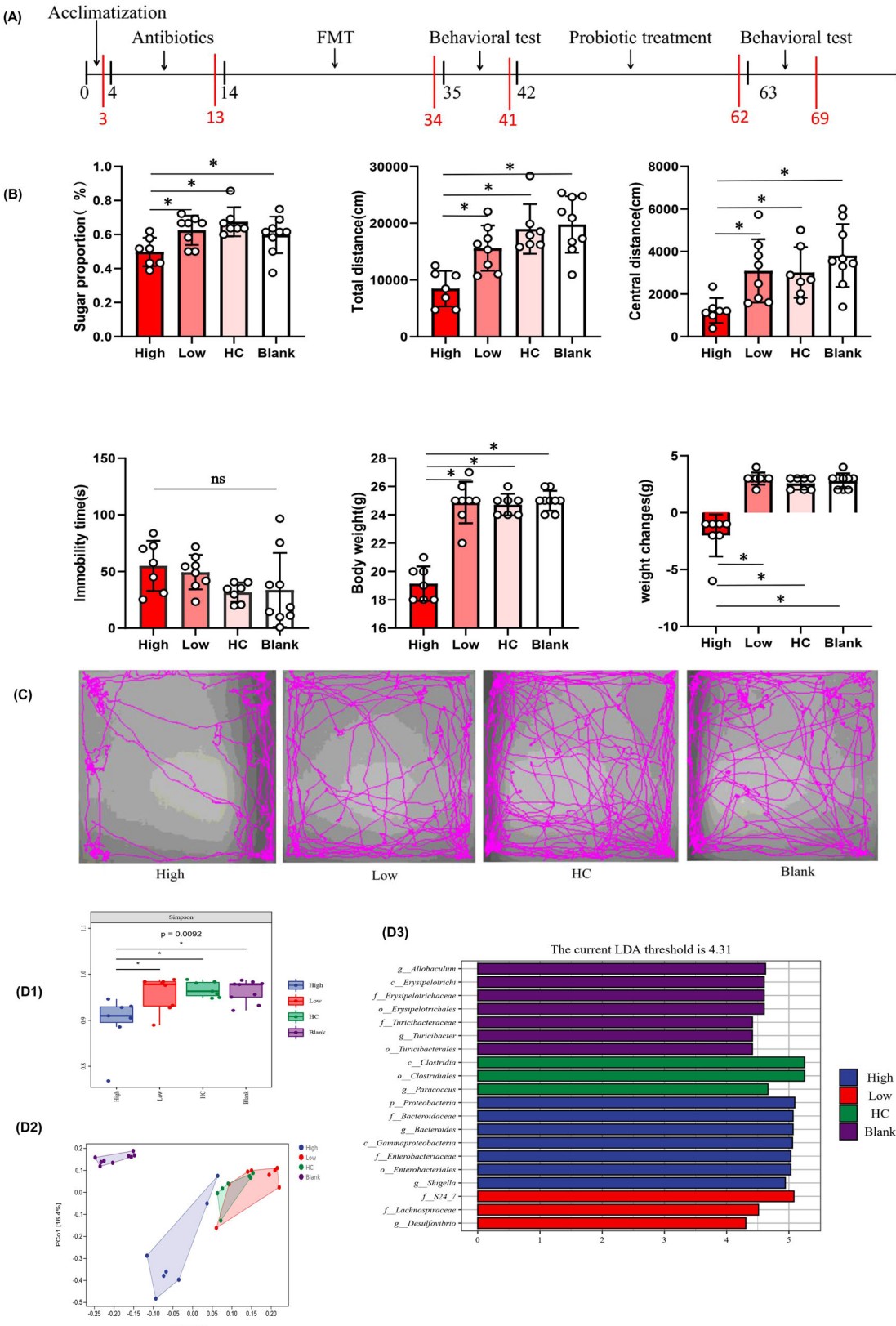

influences brain inflammation and function[30]. Innate immune pathways, including the TLR-4/NF-κB and NLRP3 pathways, play important roles in inflammatory activation[15,19]. Microbial metabolites such as SCFA are known for their anti-inflammatory functions in modulating immune cell chemotaxis, reactive oxygen species (ROS), and cytokine release[31]. In addition, we found that patients with MDD had increased

levels of inflammatory cytokines and decreased levels of permeability markers in the intestinal mucosa. This result is consistent with previous studies[27,32]. However, this study is unique relative to other work in that we measured inflammatory and permeability markers using intestinal mucosa biopsies from patients with MDD and HCs. Compared to blood plasma markers, this result was more accurate.

**Fig. 4 | Behavioral characteristics and gut microbiota composition in mouse model of inflammatory depression. A** The Schematic diagram of mouse treatment and behavioral testing. Mice were given a cocktail of antibiotics to eliminate gut microbiota and were recolonized with the fecal microbiota of Inflammatory depression patients (High-inflammatory group), Non-inflammatory depression patients (Low-inflammatory group), HCs (HC group) and normal saline (NS) (Blank group) respectively. Then High-inflammatory group mice were given the probiotics clostridium butyricum (CB) for 21 days. A series of behavioral tests were carried out 24 h after the last fecal microbiota transplantation or probiotic intervention. **B** Behavioral comparisons among recipient mice receiving the gut microbiota suspension from Inflammatory depression patients (n/mice = 7), Non-inflammatory depression patients (n/mice = 8), HCs (n/mice = 7) and normal saline (n/mice = 9). The mice in the High inflammatory group consumed fewer sucrose in the sucrose preference test (SPT) ($P = 0.012$). In the open-field test (OFT), mice in the High-inflammatory group showed decreased activity (fewer total distance traveled) ($P = 0.001$) and increased anxiety (reduced travel in the exposed center region away from the walls) ($P = 0.004$). Similarly, the duration of immobility in the tail suspension test (TST) was increased in the High-inflammatory group mice ($P = 0.061$). The body weight of mice was significantly lower and the weight change of mice in

the high inflammatory group was also smaller than that in other groups ($P = 0.001$). One-way ANOVA test for multiple comparisons with Tukey's test for post hoc corrections. Data are presented as mean values with SD. Scatter plot indicate median and error range. The upper and lower whiskers indicate median ± SD. *P*-value is marked as follows: *$P \leq 0.05$. ns indicates non-significant. **C** The movement trajectory of mice in OFT. **D1** Alpha-diversity analysis exposed that the Simpson index were lower in High inflammatory group ($n = 7$) than that in the Low-inflammatory group ($n = 8$), HC group ($n = 7$) and Blank group ($n = 9$) ($P = 0.0092$). One-way ANOVA test for multiple comparisons with Tukey's test for post hoc corrections. Data are presented as mean values with SD. Box plots indicate median and interquartile range. *P*-value is marked as follows: *$P \leq 0.05$. **D2** Beta diversity analysis uncovered a notable difference in bacterial community composition among High inflammatory group, Low-inflammatory group, HC group and Blank group as found by the PCoA plot based on Jaccard dissimilarity. **D3** LEfSe analysis showed that the relative abundance of *Bacteroidaceae, Bacteroides* was significantly higher in the High-inflammatory group; however, the amount of *Clostridia* and *Clostridiales* was significantly higher in HC group. All statistical tests are two-sided. Source data are provided as a Source Data file.

---

Although we found that patients showed low-grade inflammation, not all patients demonstrate signs of inflammation. The pathogenesis of inflammatory depression is more likely explained by the microbiota-gut-immune-brain axis[33]. We further divided patients with MDD into inflammatory and non-inflammatory depression groups based on the hs-CRP concentration in the plasma. LEfSe analysis showed increased pro-inflammatory genus *Bacteroides* and decreased SCFA-producing genus *Clostridium* in inflammatory depression compared to non-inflammatory depression and HCs. Furthermore, we identified specific microbiota at the species level using shotgun metagenomic sequencing and found that the relative abundances of 43 species were significantly lower in inflammatory depression. Most of these species belong to *Clostridium* and *Faecalibacterium*, and are correlated with hs-CRP, SCFAs, and depressive symptoms. To the best of our knowledge, this is one of the few studies to explore the characteristics of the gut microbiota in inflammatory depression. Many *Clostridium* and *Faecalibacterium* species have been demonstrated to have anti-inflammatory properties[34].

*Faecalibacterium prausnitzii* is the most abundant bacterium in the intestinal microbiota of healthy adults, representing over 5% of the total bacterial population. Recently, many studies have clearly described the abundance of *F. prausnitzii* have been linked to several human disorders, including depression[35,36]. Hao et al. showed that *F. prausnitzii* administration led to higher levels of SCFAs in the cecum and higher levels of cytokine IL-10 in the plasma, preventing the effects of CUMS on CRP and cytokine IL-6 release[36]. *F. prausnitzii* produces a microbial anti-inflammatory molecule protein (MAMP) that has been suggested to alleviate colitis in vivo and decrease the activation of NF-κB signaling[37].

*C. butyricum*, a gram-positive, spore-forming, obligate anaerobic rod bacterium, is found in the feces of 10–20% of healthy humans[38]. Previous studies have reported that CB administration reduced inflammatory cytokines, including IL-1β, IL-6, COX-2, and TNF-α, while increasing IL-10 expression levels in colon tissue[39,40]. Importantly, they both produce the SCFA butyrate[41], which was confirmed in our study by the analysis of the differential metabolic pathways of the gut microbiota. K00244 (fumarate reductase A) plays an important role in butanoate metabolism and is significantly decreased in the inflammatory depression group. Butyrate serves as a colonic fuel source, fosters immunoregulation, and promotes epithelial barrier integrity[40]. Lower levels of this genus and species have been associated with inflammatory bowel disease, autoimmune disorders, atherosclerotic cardiovascular disease, and mental health disorders including depression[32,42].

In animal study, we transplanted gut microbiota from patients with inflammatory depression, non-inflammatory depression, HCs,

and normal saline into C57BL/6J mice in the high-inflammatory, low-inflammatory, HC, and blank groups, respectively, and found that the microbial diversity and composition of the mice were similar to that of the donor. Gut microbiota of inflammatory depression mice increased the inflammatory factors TLR-4 and NLRP3 (intestinal mucosa); hs-CRP (serum); TLR-4, NF-κB, NLRP3, Caspase-1, IL-1β, and iba-1 (brain); and intestinal mucosal permeability. We also found that the high- inflammatory group showed increased depressive and anxiety-like behaviors. Gut microbiota can directly modulate the immune system, contributing to the maintenance and breakdown of immune tolerance[17]. The main components of the intestinal innate immune system include Paneth cells, dendritic cells, macrophages, neutrophils, natural killer cells, and mast cells. Most innate immune responses are mediated by PRRs, such as transmembrane surface or endosome TLRs or NLRs, which recognize MAMPs expressed by the gut microbiota. Indeed, enteric bacteria and/or MAMPs can modulate innate immune and inflammatory responses via TLRs and/or NLRs[19]. The pro-inflammatory dysbiosis in inflammatory depression patients, mainly characterized by a decrease in SCFA-producing bacteria and an increase in pathogenic bacterial strains, could promote the overactivation of the TLR-4/NF-κB and NLRP3 inflammasome signaling pathways in immune/inflammatory cells, which, in turn, could impair both the IEB and BBB while shaping both peripheral and central neurogenic/immune-inflammatory responses[15,19]. Furthermore, they activate hippocampal microglia with an increased number of activated branches[22]. Activated microglia trigger a multitude of downstream biological effects on the neuroendocrine, monoaminergic, and oxidative stress systems[23] and may subsequently lead to clinical depression[24].

TLR4s are known to cause inflammation by triggering the innate immune system. Studies have reported that TLR4/NF-κB is a key transcriptional pathway regulating the secretion of inflammatory cytokines such as IL-6, TNF-α, and IL-1β. NF-κB is activated rapidly via the MyD88-dependent pathway, which increases the transcription of downstream inflammation-related genes[43]. The NLRP3 inflammasome is a member of the NLR family and promotes innate immune defense through the maturation of inflammatory cytokines IL-1β and IL-18. NLRP3 interacts with ASC, activates Caspase-1, and produces inflammation[44]. Canonical NLRP3 activation depends on NF-κB, the activation product of the TLR-4 signaling pathway[19].

Finally, we found that CB treatment normalized gut microbiota, repaired "gut leakage," decreased peripheral and central inflammatory factors, and played an antidepressant-like role in a mouse model of inflammatory depression. CB is a strictly anaerobic gram-positive endospore-forming probiotic with acid- and heat-resistant properties that have been widely used to improve gastrointestinal function[45]. CB

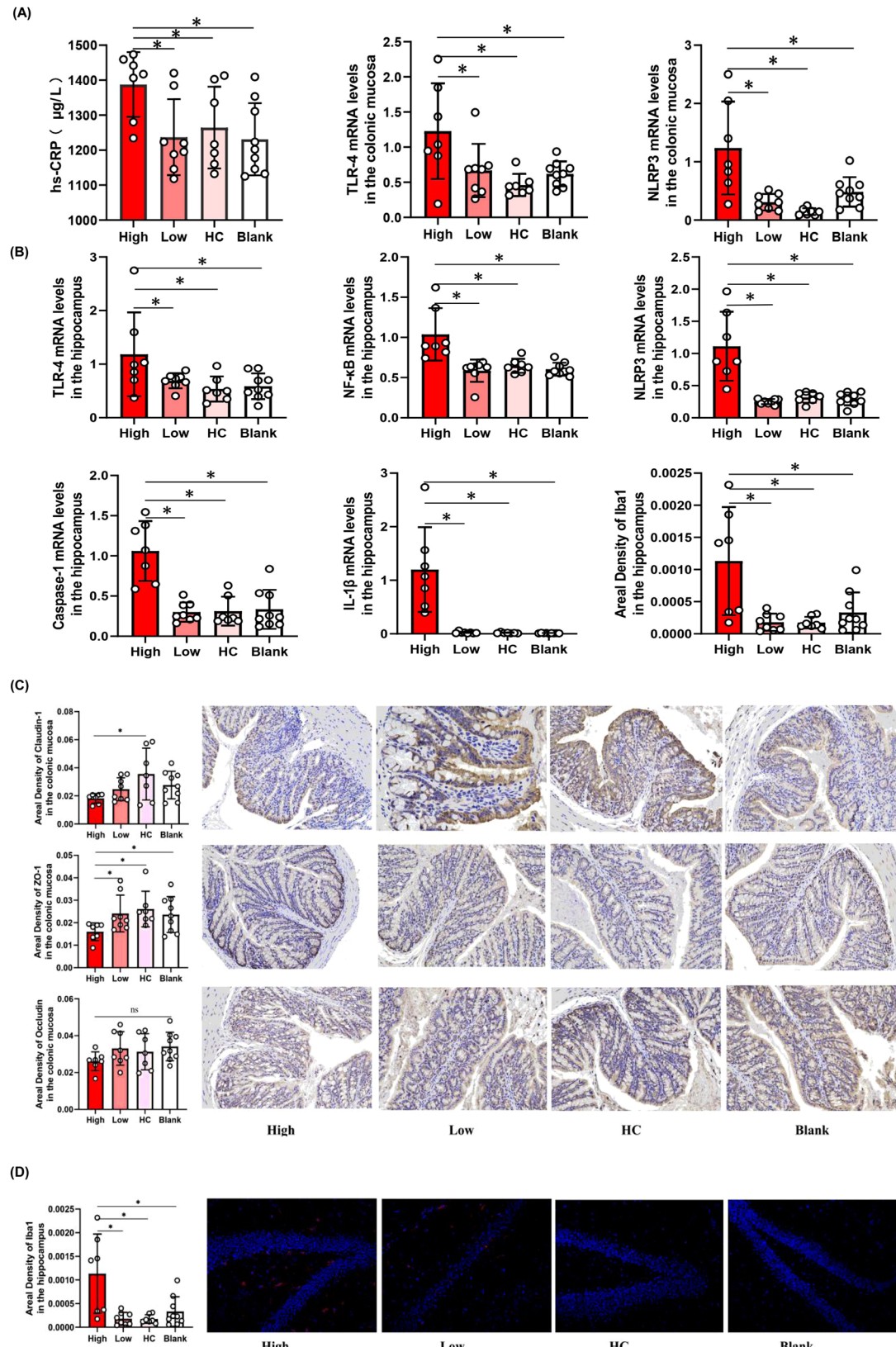

produces acetic and butyric acids, two important components of SCFAs synthesized by the gut microbiota[46]. SCFAs are neuroactive bacterial metabolites that can translocate from the gut lumen to systemic circulation and subsequently cross the BBB[47]. SCFAs are considered exhibiting anti-inflammatory properties under certain conditions, thereby mitigating LPS-induced immune responses[48]. Moreover, SCFAs may have beneficial effects on the integrity of the intestinal barrier and the BBB. Butyrate can be systemically disseminated and detected in the rat brain[31]. Butyrate in the rat brain can exert neuroprotective effects against neurodegenerative disorders

**Fig. 5 | Elevated inflammatory factors in the High inflammatory group. A** The expression of TLR-4 ($P = 0.006$), NLRP3 ($P < 0.001$) in the intestinal mucosa and the concentration of hs-CRP ($P = 0.028$) in the serum were elevated in the High inflammatory group ($n = 7$) compared to the Low-inflammatory group ($n = 8$), HC group ($n = 7$), Blank group ($n = 9$). **B** The expression of TLR-4 ($P = 0.023$), NF-κB ($P < 0.001$), NLRP3 ($P < 0.001$), Caspase-1 ($P < 0.001$) and the area density of Iba1 ($P = 0.001$) in the brain were increased in the High inflammatory group. **C** The amount and area density of permeability biomakers such as Claudin-1 ($P = 0.043$), ZO-1 ($P = 0.003$) were decreased in the High inflammatory by ELISA and immunohistochemistry (×400 magnificent). The area density of Occludin tended to decrease, but there was no statistical difference ($P = 0.254$). **D** The amount and branch of microglia in hippocampus were increased in the High inflammatory group. One-way ANOVA test for multiple comparisons with Tukey's test for post hoc corrections. Data are presented as mean values with SD. Scatter plot indicate median and error range. The upper and lower whiskers indicate median± SD. *P*-value is marked as follows: *$P ≤ 0.05$. ns indicates non-significant. All statistical tests are two-sided. Source data are provided as a Source Data file.

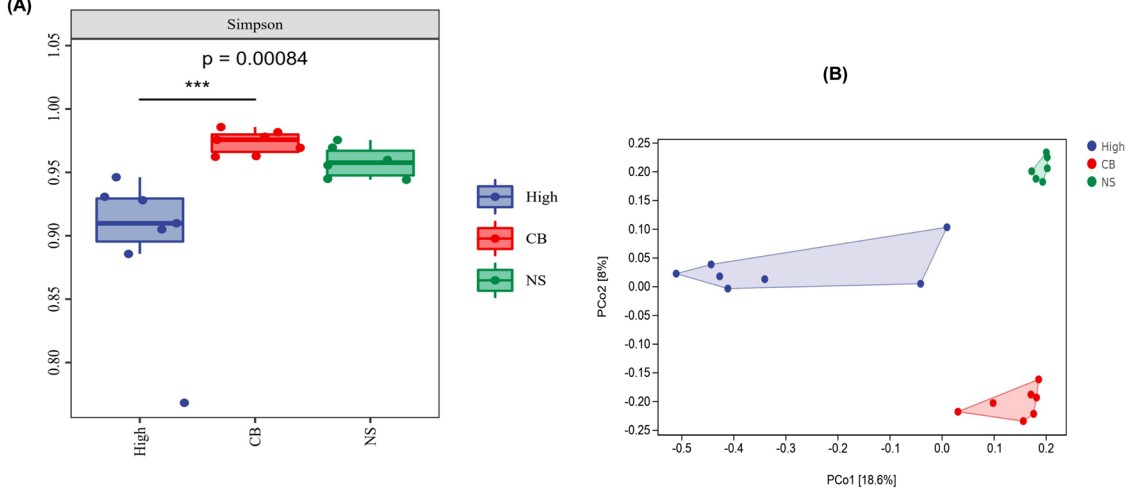

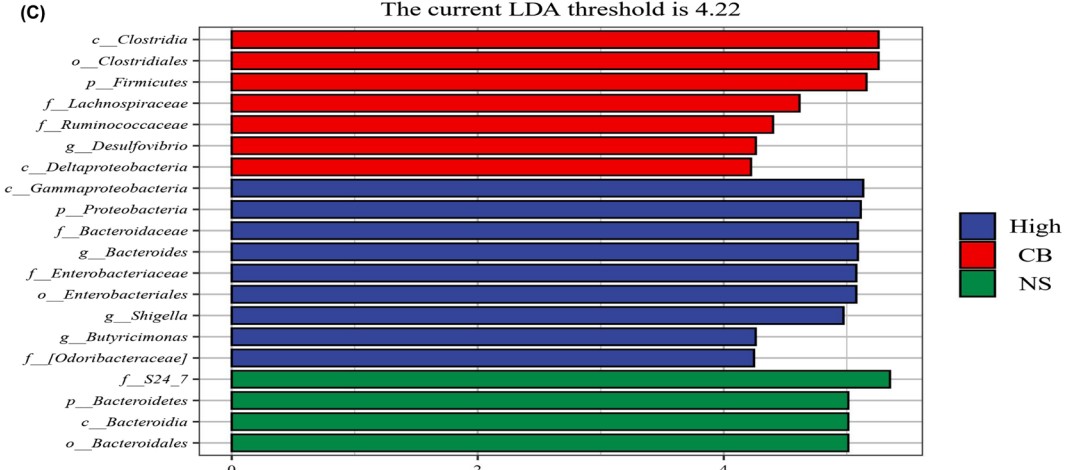

**Fig. 6 | Effects of probiotics (Clostridium butyricum) on gut microbiota of the mouse model of inflammatory depression. A** Alpha-diversity analysis exposed that CB increased the Simpson index (n/High inflammatory group = 7, n/CB group = 7, n/NS group = 6). One-way ANOVA test for multiple comparisons with Tukey's test for post hoc corrections. Data are presented as mean values with SD. Box plots indicate median and interquartile range. *P*-value is marked as follows: ***$P ≤ 0.001$. **B** Beta diversity analysis uncovered the bacterial community composition was notably different between High inflammatory group ($n = 7$) and CB group ($n = 7$). **C** LEfSe analysis showed that the abundance of Clostridia and Clostridiales were increased and that of Bacteroidaceae, *Bacteroides* were decreased in CB group ($n = 7$). All statistical tests are two-sided. Source data are provided as a Source Data file. CB Clostridium butyricum, NS Normal saline.

and improve behavioral deficits via anti-inflammatory responses[49], suggesting that it plays an important intermediary role in the alteration of neurobiological functions. In addition, CB may be involved in the regulation of the intestinal microbiota by decreasing pro-inflammatory bacteria and increasing beneficial bacteria[39]. Moreover, we found a higher abundance of *Clostridium* and a lower abundance of *Bacteroides* at the genus level in the CB group, which may partially suggest CB colonization. In comparison to patients without CB treatment, the α-diversity index of gut microbiota was more abundant in patients who received CB[50]. Mao et al. showed that CB increased the abundance of *Bifidobacterium*, *Lactobacillus*, and *Lactococcus* species in the gut microbiome[51]. Furthermore, Viktoriya et al. found that some Bifidobacterium and Lactococcus species also have antidepressant effects in a single-center, double-blind, placebo-controlled, pilot randomized clinical trial[52]. Thus, CB can modulate inflammatory responses and improve depressive symptoms by increasing SCFAs or regulating the gut microbiota.

Our study has some limitations. First, our clinical sample size was small, particularly for the intestinal mucosa. In our study, intestinal mucosa was collected from only 12 subjects, including

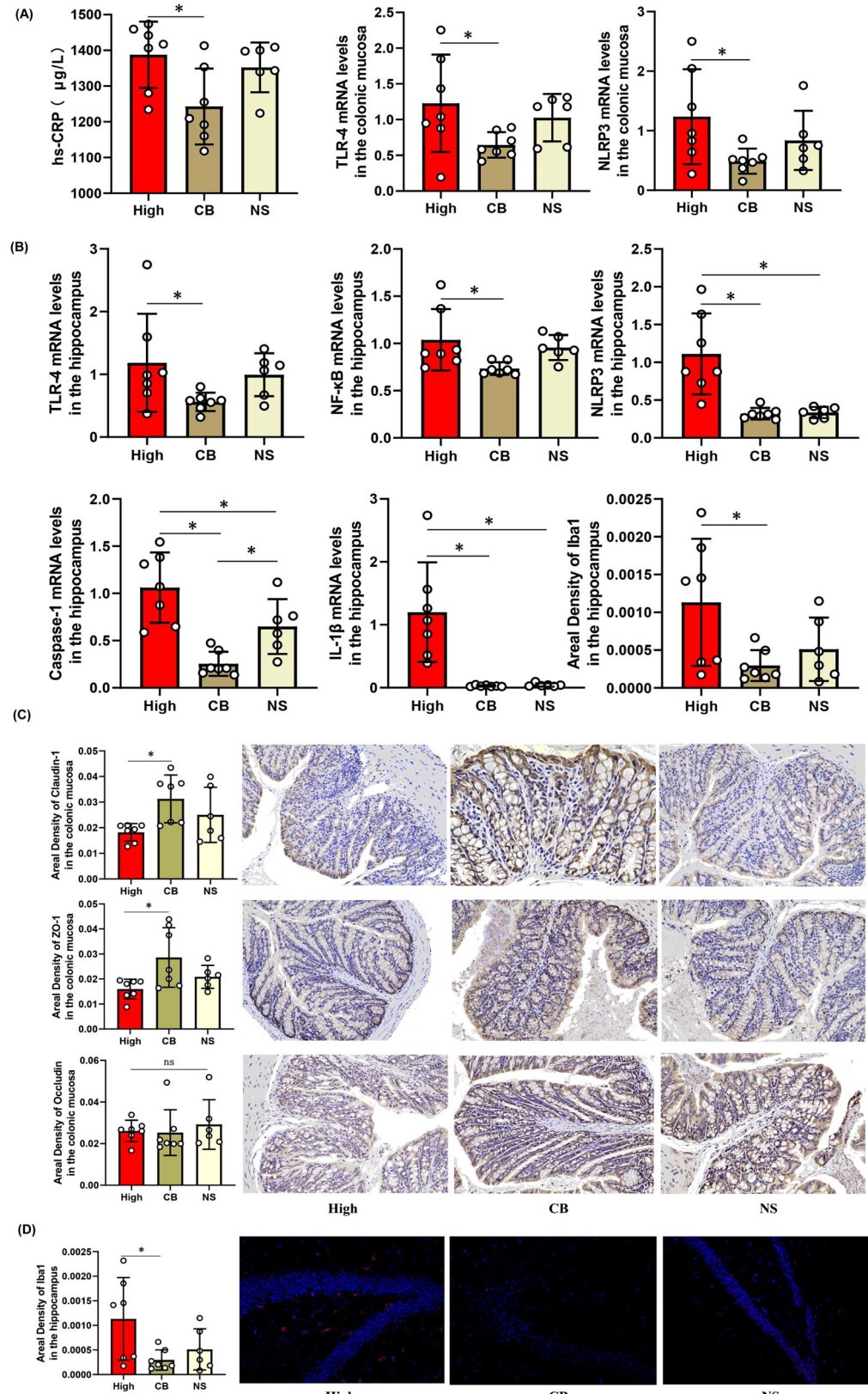

six patients with MDD and six healthy subjects. Furthermore, it was challenging to recruit patients without gastrointestinal symptoms who are willing to undergo sigmoidoscopy or sigmoid biopsy. Similarly, the sample size ($n = 6$) in the NS group was small in animal experiments. In addition, we identified specific microbiota at the species level using shotgun metagenomic sequencing. However, the efficacy of SCFAs producing CB against inflammatory depression has only been verified in animal studies without a clinical trial.

In summary, we found that the gut microbiota of inflammatory depression patients had specific characteristics, including increased pro-inflammatory genera and decreased SCFA-producing genera, from

**Fig. 7 | Effects of probiotics (CB) on inflammatory factors of the mouse model of inflammatory depression. A** The CB decreased the expression of TLR-4 ($P = 0.018$), NLRP3($P = 0.023$) in the intestinal mucosa and the concentration of hs-CRP in the serum ($P = 0.024$). (n/High inflammatory group = 7, n/CB group = 7, n/NS group = 6). **B** The CB decreased the expression of TLR-4($P = 0.048$), NF-κB($P = 0.041$), NLRP3($P < 0.001$) and Caspase-1($P < 0.001$) in the brain. **C** The amount and area density of permeability biomakers such as Claudin-1($P = 0.030$), ZO-1($P = 0.026$) were increased in the CB group by ELISA and immunohistochemistry (×400 magnificent).The area density of Occludin tended to increase, but there was no statistical difference ($P = 0.750$). **D** The area density, amount and branch of microglia in hippocampus were decreased in the CB group ($P = 0.034$). One-way ANOVA test for multiple comparisons with Tukey's test for post hoc corrections. Data are presented as mean values with SD. Scatter plot indicate median and error range. The upper and lower whiskers indicate median ± SD. *P*-value is marked as follows: *$P \leq 0.05$. ns indicates non-significant. All statistical tests are two-sided. Source data are provided as a Source Data file.

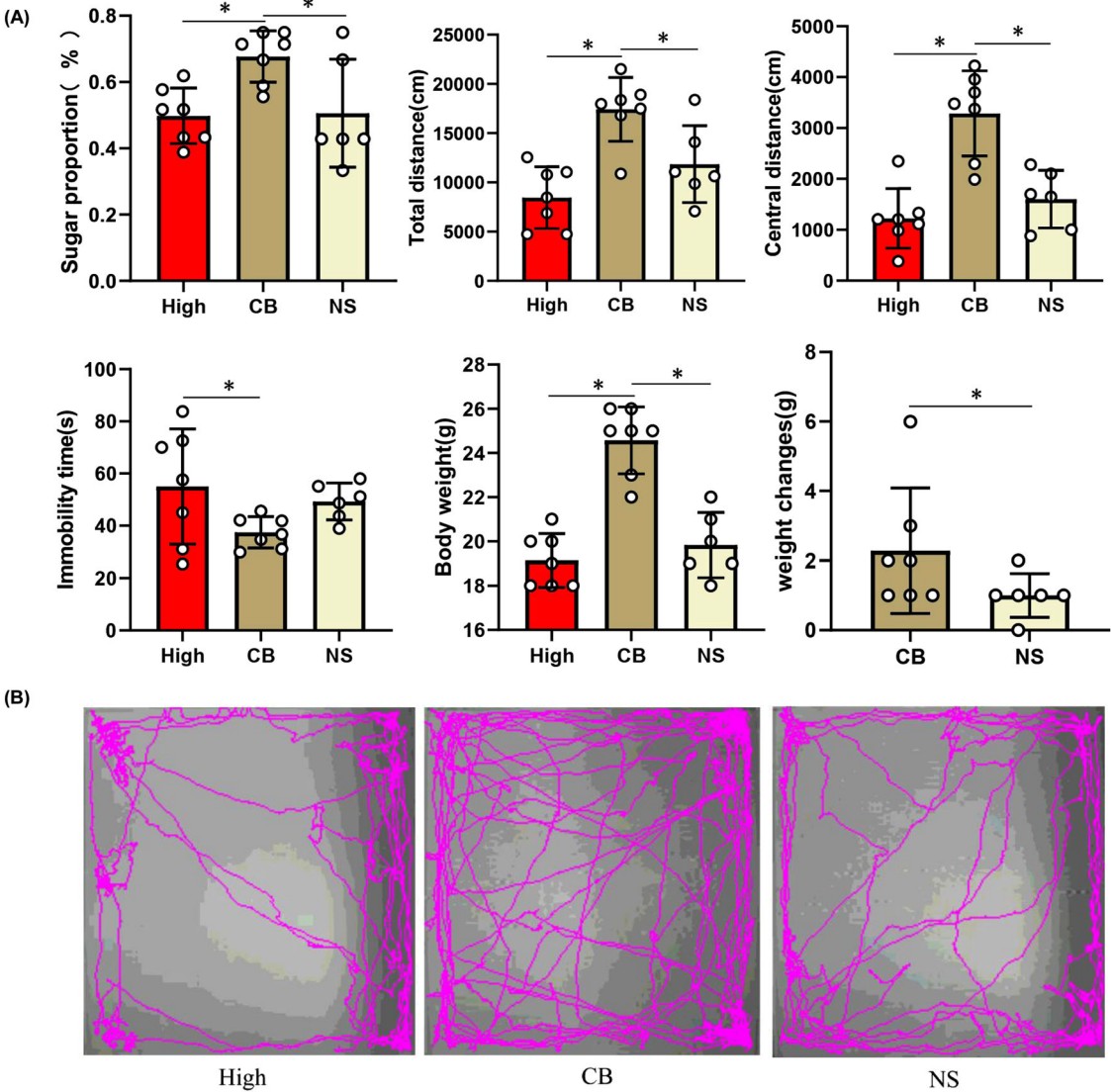

**Fig. 8 | Effects of probiotics (CB) on depression-like behaviors of the mouse model of inflammatory depression. A** The mice in the CB group ($n = 7$) consumed more sucrose in the SPT than that in High-inflammatory group ($n = 7$) and NS group ($n = 6$) ($P = 0.013$). In the OFT, mice in the CB group showed increased activity (longer total distance traveled) ($P = 0.001$) and decreased anxiety (induced travel in the exposed center region away from the walls) ($P < 0.001$). Similarly, the duration of immobility in the TST was decreased in the CB group mice ($P = 0.048$). The body weight of mice was significantly increased and the weight change of mice was also more in the CB group ($P < 0.001$). One-way ANOVA test for multiple comparisons with Tukey's test for post hoc corrections. Data are presented as mean values with SD. Scatter plot indicate median and error range. The upper and lower whiskers indicate median ± SD. *P*-value is marked as follows: *$P \leq 0.05$. ns indicates non-significant. **B** The movement trajectory of mice in OFT. All statistical tests are two-sided. Source data are provided as a Source Data File.

humans to mice. Gut microbiota-derived inflammatory processes are involved in neuroinflammation in patients with inflammatory depression, and the TLR-4/NF-κB and NLRP3 inflammasome signaling pathways, at least in part, mediate this interaction. Our study suggests that gut microbiota is a promising therapeutic target for inflammatory depression and provides a scientific basis for future interventional studies.

## Methods
### Clinical study
**Participants and sample collection procedures.** 85 first-episode, drug-naïve patients with MDD ages of 18–55 were recruited from the Department of Psychiatry at the First Hospital of Shanxi Medical University between December 2019 and July 2021. All patients met the criteria for MDD according to the fourth edition of the Diagnostic and

Statistical Manual of Mental Disorders-Fourth Edition (DSM-IV). Depression and anxiety severity were evaluated using HAMD-17[53] and HAMA[54]. The Mini International Neuropsychiatric Interview (MINI) was used to screen for manic or hypomanic episodes and other psychiatric disorders[55]. Individuals were excluded if they had other psychiatric disorders, any somatic illness, a history of alcohol and/or drug abuse, or had taken antibiotics, probiotics, prebiotics, or synbiotics within the past three months. An advertisement was used to recruit healthy controls (HCs) from the local community and universities; 85 HCs participated in the study. The subjects had no physical or somatic illnesses. In addition, none of the HCs used antibiotics, probiotics, prebiotics, or synbiotics within the previous three months. Written informed consent was obtained from all the participants. In addition, minor participants (age less that 18 years) were involved in this study, the informed consent was obtained from parents of these participants. This study was approved by the Research Ethics Review Board of the First Hospital of Shanxi Medical University, Taiyuan, China. This study was registered in the Chinese Clinical Trial Registry (ChiCTR1900025175). We confirm that none of the pre-specified outcomes are reported and we are only reporting baseline data and nothing else of the clinical trials. Meanwhile, we obtained the ethical approval for interim reporting of this baseline data.

Patients with MDD were divided into inflammatory and non-inflammatory depression groups based on peripheral blood hs-CRP concentration. The cut-off point[8] was the second tertile (66.7%) of hs-CRP in the peripheral blood of HCs.

### Collection of fecal, blood, and colon biopsy samples
Stool samples were collected after the participants completed the questionnaire assessments; the samples were immediately frozen at −80 °C before a DNA extraction kit (Imunobio Co. Ltd., Shenzhen, China) was utilized for DNA extraction. DNA was extracted from stool samples using a Stool GenDNA kit (CW Biotech Co., Beijing, China). Ten milliliters of blood were drawn from participants using normal aseptic techniques. Plasma was separated by centrifugation at 3500 × g at 4 °C for 10 min and stored at −80 °C for SCFA, IL-1β, IL-6, IL-10, TNF-α, and hs-CRP measurement. Subjects were scheduled for a colonoscopy, and 8 mucosal biopsy samples were taken from the distal sigmoid colon of each subject; 4 mucosal biopsy samples were stored in −80 °C refrigerator for inflammatory factor measurement by protein chips and ELISA, and the remaining 4 were placed in formalin for the detection of intestinal permeability.

### Microbiota analysis
The microbiota composition of 85 patients with MDD and 85 HCs was investigated using 16S RNA sequencing. Gut microbiota from 20 patients with inflammatory depression and 20 HCs were randomly selected for metagenomic shotgun sequencing. Only 16S ribosomal RNA sequencing was used in the animal experiments, as described in the online supplemental materials and methods.

### Inflammatory and permeability marker measurements
Plasma cytokines IL-1β, IL-6, IL-10, TNF-α, and hs-CRP were detected by ELISA. To prevent the deviation of the data from the detection, we performed three repeated measurements of the same plasma sample and then took the average value of the three values. An antibody cytokine array (RayBiotech, GSH-INF-3 and GSH-GF-1) consisting of 80 inflammatory factors was used to screen for differential intestinal mucosal inflammatory factors between patients with MDD and HCs. Immunohistochemistry and microscopic analyses of the human intestinal mucosa ($n = 6$ subjects with MDD and $n = 6$ HCs) were performed to assess the integrity of the tight junction proteins zonula occludens 1 (ZO-1) (Antibodies: Servicebio, China, dilution 1:500, Rabbit, Lot, GB111981)[56], Claudin-1 (Antibodies: Servicebio, China, dilution 1:500, Mouse, Lot, GB12032. https://www.servicebio.cn/

goodsdetail?id=14247), and Occludin (Antibodies: Servicebio, China, dilution 1:500, Rabbit, Lot, GB111401)[57] markers, as described in the online supplemental materials and methods.

To further clarify the inflammatory pathway and quantify intestinal permeability, ELISA was used to measure inflammatory factors such as TLR-4, NF-κB, Caspase-1, and NLRP3 inflammasomes, in addition to mucosal permeability markers such as Claudin-1, ZO-1, and Occludin.

### Targeted metabolomic analysis of SCFAs in plasma
Targeted SCFA panel profiling was performed using the metabolons in plasma samples from the participants. 7 SCFAs including acetic acid (C2), propionic acid (C3), isobutyric acid (C4), butyric acid (C4), iso-valeric acid (C5), valeric acid (C6), and caproic acid (hexanoic acid, C7) were quantified by gas chromatography-mass spectrometry (GC-MS) as previously published[58] and described in the online supplemental material and methods.

### Statistical analyses
IBM SPSS Statistics for Windows, version 23.0 (IBM Corp., Armonk, NY, USA) was used for statistical analyses. Two-sample t-tests and one-way analysis of variance (ANOVA) were performed to determine the differences among the measured data. Data were compared using the $x^2$ test. Pearson or Spearman correlation analyses were performed to determine the correlation coefficient, and FDR correction was used. Data from the protein chips were analyzed using moderated t-statistics and corrected using the Benjamini−Hochberg method. The threshold of statistical significance was set at $P < 0.05$ (two-tailed). No statistical method was used to predetermine sample size.

### Animal study
The animal experiments were approved by Ethical Review Committee of Experimental Animal Welfare of the First Hospital of Shanxi Medical University. The schematic diagram of mouse treatment and behavioral testing is showed in Fig. 4A.

### Animal housing
Male C57BL/6 J mice ($n = 44$) were obtained from SPF Biotechnology Co., Ltd. (Beijing) (six weeks of age; 4–5 per cage). Mice were maintained in a temperature-controlled (21–23 °C) and relative humidity-controlled (50–60%) environment with a 12/12-h light–dark cycle. The mice were provided with standard chow and autoclaved water ad libitum.

### Mouse model for inflammatory depression
We performed FMT experiments to determine whether inflammatory depression-related behavioral phenotypes were linked to disturbed gut microbiota. The global gut microbial phenotypes of the randomly selected subset of samples used for these FMT experiments were representative of their full population distributions. Ten fecal samples were randomly selected from patients with inflammatory depression, patients with non-inflammatory depression, and HCs for microbiota transplantation according to the methods described in a previous study[59]. Briefly, feces were suspended in glycerin-PBS and centrifuged immediately after collection. Supernatants containing microbiota from patients or controls were mixed and used for transplantation. The gut microbiome of the mice was depleted by oral gavage with a cocktail of antibiotics as described in previous studies after acclimatization for three days[60]; the procedure is shown in the supplementary information. After the last gavage of antibiotics (10 d), mice were randomly classified into 4 groups: high-inflammatory group ($n = 20$), low-inflammatory group ($n = 8$), HC group ($n = 7$), and blank group ($n = 9$), received an oral gavage of the microbiota suspension of patients with inflammatory depression, patients with non-

inflammatory depression, HCs, and NS (10 μL/g body weight), respectively, for 3 weeks to reconstruct gut microbiota.

## Probiotic intervention
Seven mice with inflammatory depression were euthanized after FMT, and the rest were randomly divided into two groups: probiotic group ($n = 7$) and NS group ($n = 6$). The two groups were administered with CB ($6.3 \times 10^6$ CFU/ml) and NS by gavage (0.2 mL/mouse) from 8:00 a.m. to 9:00 a.m. continuously for 3 weeks.

## Behavioral testing
All behavioral tests were conducted 24 h after FMT or probiotic intervention. The results were analyzed using a computerized video-tracking system or scored manually. SPT, OFT, and TST were used to evaluate anhedonia, anxiety, and depression in mice respectively as described in a previous study[61–64] and as described in the online supplemental material and methods.

## Weighing
All the mice were weighed after FMT and probiotic intervention. In general, the weight of the mice correlates with appetite.

## Blood, fecal, and tissue sample collection and measurement
After the behavioral test, the mice were anesthetized with 10% chloral hydrate (0.1 mL/10 g), and blood was obtained from the orbital venous plexus. These blood samples were centrifuged by $1000 \times g$ for 20 min to collect the serum samples, which were immediately frozen at −80 °C for biochemical assay. ELISA was used to measure the concentration of high-sensitivity C-reactive protein (hs-CRP). We performed three repeated measurements of the same plasma sample during ELISA and then averaged the three values. Stool samples were collected from the colon, and the microbiota composition was analyzed using 16S rRNA sequencing, as described in the online supplemental material and methods section. The colon tissues were dissected from euthanized mice, and the expression of TLR-4 and NLRP3 was detected by quantitative polymerase chain reaction (qPCR), while the permeability biomarkers Claudin-1 (Antibodies: Servicebio, China, dilution 1:500, Mouse, Lot, GB12032. https://www.servicebio.cn/goodsdetail?id=14247), ZO-1(Antibodies: Servicebio, China, dilution 1:500, Rabbit, Lot, GB111981)[56], and Occludin (Antibodies: Servicebio, China, dilution 1:500, Rabbit, Lot, GB111401)[57], were measured by immunohistochemistry. The hippocampus was then dissected from the mice's brain and stained with iba-1 antibody (Servicebio, China, dilution 1:800, Rabbit, Lot, GB113502. https://www.servicebio.cn/goodsdetail?id=6764) to assess the amount and morphology of microglia, while the expression of TLR-4, NF-κB, NLRP3, and Caspase-1 was detected by qPCR.

## Statistical analyses
IBM SPSS version 23.0 was used for the statistical analyses. Results are expressed as mean ± standard error of the mean (SEM). One-way ANOVA and post hoc tests were performed to determine the differences between the measurement data. Pearson or Spearman correlation analysis was used to determine the correlation coefficients. The threshold of statistical significance was set at $P < 0.05$ (two-tailed).

## Reporting summary
Further information on research design is available in the Nature Portfolio Reporting Summary linked to this article.

## Data availability
The raw data of gut microbiota generated in this study have been deposited in the National Center of Biotechnology Information (NCBI) database under accession code PRJNA1081663 (16S ribosomal RNA sequences in human), PRJNA1083304 (Metagene sequences in human), PRJNA1082123 (16S ribosomal RNA sequences in mice) [https://www.ncbi.nlm.nih.gov/sra]. The SCFAs data have been deposited to the ProteomeXchange Consortium (https://proteomecentral.proteomexchange.org) via the iProX partner repository with the dataset identifier PXD050388. All data are subject to open access. The source data are provided in this study. Source data are provided with this paper.

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

## Acknowledgements

We thank all the participants who provided clinical data, blood, stool, and intestinal mucosa samples. We are also grateful to the Key Laboratory of Cellular Physiology (Shanxi Medical University) and Shanxi Key Laboratory of Artificial Intelligence Assisted Diagnosis and Treatment for Mental Disorder (First Hospital of Shanxi Medical University) for providing us with animal experiments. This study was financially supported by the National Natural Science Youth Fund Project (82201691) and Youth Scientific Research Project of the Shanxi Basic Research Program (20210302124193) led by L.P.H. and National Natural Science Foundation of China (81471379) led by Z.K.R.

## Author contributions

Kerang Zhang and Ning Sun designed experiments. Mingxue Gao, Yanyan Zhang, Aixia Zhang, Chunxia Yang, Gaizhi Li, Xinrong Li, Zhifen Liu participated in the collection and analysis of clinical data, stool samples, and blood samples of all subjects. Jizhi Wang and Junyan Wang performed animal experiments. Sha Liu and Lixin Liu instructed the animal experiments. Penghong Liu and Zhifen Liu analyzed the data and wrote the manuscript.

## Competing interests

The authors declare no competing interests.
