## [Peer Review File · Nature Communications]

Immunoregulatory role of the gut microbiota in inflammatory depressionREVIEWER COMMENTS

Reviewer #1 (Remarks to the Author):

This is an interesting study examining the potential relationship between gut microbiota abnormalities, and the existence of a pro-inflammatory profile in depression. It combines a mice and human study.

The study is very clearly-written and the findings are very interesting and of potential relevance.

I have a couple of relatively minor concerns:

The sample size is relatively low (n=6 for some markers), so I think, they should highlight it (a little bit more) in the limitations section.

I think it's interesting also to highlight that patients of the here reported cohort were not overweight. Previous literature has reported the existence of increased levels of the determined and overweight (so called, metabolic depression). Patients of the here reported cohort had no other somatic conditions, and were not overweight, this should be highlighted and include a short sentence with literature.

They could also support their findings by a recently published paper: DOI: 10.1001/jamapsychiatry.2023.1817

Reviewer #2 (Remarks to the Author):

The manuscript by Liu et al., describes the role of the gut microbiome in major depressive disorder with or without inflammation and uses a translational approach to do so. The study is highly interesting and benefits from the use of mice and humans but there are a number of major issues that need to be addressed:

1. There are numerous methodological details that are missing or not described in enough detail. These include n numbers for various readouts and whether or not gender was considered as a factor in the clinical trials. The overall groups are do not vary but gender differences in MDD, as well as many of the readouts are known. How many of each gender are in the inflammation vs non-inflammation groups. Did the authors perform appropriate multiple corrections when assessing multiple measures from the same plasma samples etc?
2. How were the FMT samples selected for the mouse study - what criteria were employed? Similar questions arise for the selection of the shotgun samples and the very low number for the colon IHC.
3. In the animal studies, it is stated that behavioral tests were performed 24h after FMT - but according to the supplementary methods FMT was performed for 3 weeks (which would be Day 14 to Day 35 in the main figure) - however the behavioral tests were performed from day 35 to 42 (a similar question arises for the probiotic timeline).
4. The methods and the supplementary methods describe the tail suspension test but the results refer to the forced swim test - which is it? This would change the scoring of the tests as they differ.
5. The low patient number is mentioned as a limitation - but this is equally true of the animal study as some groups are only n=6, which also makes the vast number of correlations performed of a low power.
6. Why were low inflammatory patients not included in the shotgun metagenomic study to determine whether they are more similar to controls or to the high inflammation group?

7. The study was performed in male mice - did they receive FMT from female patients or only males?

8. Given that the high FMT mouse group show decreased locomotor activity, this compromises the TST (or FST) findings as low activity may be the reason for the increased immobility (and reduced time in the center of the open field). Similarly, the body weight reduction may be reflected in the sucrose test. Did this group consume a overall fluid volume and was side-preference taken into consideration?

9. How was the hippocampus dissected and then the Iba1 IHC performed? Was this via a cryostat or vibratome? or if dissected, which part of the hippocampus was used and how was the area assessed controlled for?

10. Does the probiotic (CB) have effects on sucrose preference or other parameters per se - or are these related to the high MDD FMT?

Thanks to reviewers for reading our manuscript and making valuable suggestions. We have made the revision according to the comments.

Reviewer #1 (Remarks to the Author):

This is an interesting study examining the potential relationship between gut microbiota abnormalities, and the existence of a pro-inflammatory profile in depression. It combines a mice and human study.

The study is very clearly-written and the findings are very interesting and of potential relevance.

I have a couple of relatively minor concerns:

The sample size is relatively low (n=6 for some markers), so I think, they should highlight it (a little bit more) in the limitations section.

Response: Thank you for this suggestion. We have added “In our study, intestinal mucosa was only collected from 12 subjects, including 6 MDD patients and 6 healthy subjects” in the limitations section. This is on page 20 of the manuscript, lines 592 to 594.

I think it’s interesting also to highlight that patients of the here reported cohort were not overweight. Previous literature has reported the existence of increased levels of the determined and overweight (so called, metabolic depression). Patients of the here reported cohort had no other somatic conditions, and were not overweight, this should be highlighted and include a short sentence with literature.

Response: Thank you for this suggestion. We have added “Patients of this reported cohort had no other somatic conditions and were not overweight” in the part of clinical study in result section. This is on page 7 of the manuscript, lines 191 to 192.

They could also support their findings by a recently published paper: doi: 10.1001/jamapsychiatry.2023.1817

Response: Thank you for providing the reference, which also support our findings. We have added “Furthermore, Viktoriya et al. found that some Bifidobacterium species and Lactococcus species also has antidepressant effects through a single-center, double-blind, placebo-controlled pilot

randomized clinical trial” and cited this reference in discussion section. This is on page 20 of the manuscript, lines 586 to 589.

Reviewer #2 (Remarks to the Author):

The manuscript by Liu et al., describes the role of the gut microbiome in major depressive disorder with or without inflammation and uses a translational approach to do so. The study is highly interesting and benefits from the use of mice and humans but there are a number of major issues that need to be addressed:

1. There are numerous methodological details that are missing or not described in enough detail. These include n numbers for various readouts and whether or not gender was considered as a factor in the clinical trials. The overall groups are do not vary but gender differences in MDD, as well as many of the readouts are known. How many of each gender are in the inflammation vs non-inflammation groups. Did the authors perform appropriate multiple corrections when assessing multiple measures from the same plasma samples etc?

Response: Thank you for your suggestions. Firstly, we have supplemented all n numbers in the manuscript. Secondly, we analyzed the ratio of males to females among the three groups, but there was no statistical difference, so we believe that gender had no effect on the results. Thirdly, we perform multiple corrections when assessing multiple measures from the same plasma samples.

2. How were the FMT samples selected for the mouse study - what criteria were employed? Similar questions arise for the selection of the shotgun samples and the very low number for the colon IHC.

Response: Thank you for your suggestions. We selected the FMT samples and the shotgun samples by systematic random sampling. However, because most of the subjects were unwilling to undergo colonoscopy, colonic mucosa was collected from only 12 subjects containing 6 MDD patients and 6 healthy subjects for IHC.

3. In the animal studies, it is stated that behavioral tests were performed 24h after FMT - but according to the supplementary methods FMT was performed for 3 weeks (which would be Day 14 to Day 35 in the main figure) - however the behavioral tests were performed from day 35 to 42 (a similar question arises for the probiotic timeline).

Response: Thank you for this suggestion. It was a mistake on our part that we did not clearly mark the end date, which should have been the day 34. We have made corrections all mistakes on the flow chart. This is on page 37 of the manuscript, Fig 7-A.

4. The methods and the supplementary methods describe the tail suspension test but the results refer to the forced swim test - which is it? This would change the scoring of the tests as they differ.

Response: Thank you for this suggestion. It was our mistake and we have modified the forced swim test into the tail suspension test in the result. They are on page 13 and 16 of the manuscript, lines 382 and 470. We also made modifications in the captions of figure 7 and 13.

5. The low patient number is mentioned as a limitation - but this is equally true of the animal study as some groups are only n=6, which also makes the vast number of correlations performed of a low power.

Response: Thank you for this suggestion. This is indeed a limitation and we have added "Similarly, the sample size (n=6) in the NS group was small in the animal experiment" in the limitation section. This is on page 21 of the manuscript, lines 595 to 596.

6. Why were low inflammatory patients not included in the shotgun metagenomic study to determine whether they are more similar to controls or to the high inflammation group?

Response: It is a good question. We chose to perform metagenome shotgun sequencing in inflammatory depression group and healthy subjects in order to further identify the different strains in inflammatory depression patients and provide evidence for probiotic therapy in our animal experiments and clinical practice in the future. However, we have found differences in gut microbiota between inflammatory depression and non-inflammatory depression by 16S rRNA gene sequencing, so further metagenome shotgun sequencing for non-inflammatory depression is not necessary. Another reason is that metagenome shotgun sequencing is relatively expensive.

7. The study was performed in male mice - did they receive FMT from female patients or only males?

Response: It is a good question. Patients were randomly selected for FMT and male mice received FMT from female patients.

8. Given that the high FMT mouse group show decreased locomotor activity, this compromises the TST (or FST) findings as low activity may be the reason for the increased immobility (and reduced time in the center of the open field). Similarly, the body weight reduction may be reflected in the sucrose test. Did this group consume a overall fluid volume and was side-preference taken into consideration?

Response: It is a good question. In fact, the intaking fluid volume did affect the body weight of the mice. During the experiment, we found that the total fluid intake of the mice in the High-inflammatory group was slightly less than that of the other groups, but the difference was not statistically significant ($p > 0.05$). In addition, relative to body weight, the weight of fluid intake is small, so it had little effect on body weight. This is a good suggestion, and we will pay attention to these details in future experiments.

9. How was the hippocampus dissected and then the Iba1 IHC performed? Was this via a cryostat or vibrotome? or if dissected, which part of the hippocampus was used and how was the area assessed controlled for?

Response: Thank you for this suggestion. The hippocampus was dissected and then the Iba1 IHC performed via a cryostat. After dissected, the whole hippocampal tissue was sliced, and we tried to select the dentate gyrus of the hippocampi for observation.

10. Does the probiotic (CB) have effects on sucrose preference or other parameters per se - or are these related to the high MDD FMT?

Response: It is a good question. After probiotic (CB) intervention, the sucrose consumption of High-inflammatory mice in the SPF was significantly increased, the total distance traveled in the OFT was longer, duration of immobility in the TST was decreased and the weight was significantly increased. However, there was no significant change in the NS group, suggesting that the probiotics (CB) can alleviate depression-like behaviors of High-inflammatory mice.

REVIEWER COMMENTS

Reviewer #1 (Remarks to the Author):

This is an interesting study examining the potential relationship between gut microbiota abnormalities, and the existence of a pro-inflammatory profile in depression. It combines a mice and human study. The study is very clearly-written and the findings are very interesting and of potential relevance. Authors have properly addressed my remarks and the manuscript has improved, compared to the previous version.